# Current Progress towards the Integration of Thermocouple and Chipless RFID Technologies and the Sensing of a Dynamic Stimulus

**DOI:** 10.3390/mi11111019

**Published:** 2020-11-20

**Authors:** Kevin Mc Gee, Prince Anandarajah, David Collins

**Affiliations:** 1School of Biotechnology, Dublin City University, Dublin 9, Ireland; david.collins@dcu.ie; 2The National Centre for Sensor Research (NCSR), Research & Engineering Building, Dublin City University, Dublin 9, Ireland; 3Photonics Systems and Sensing Laboratory, School of Electronic Engineering, Dublin City University, Dublin 9, Ireland; prince.anandarajah@dcu.ie; 4School of Electronic Engineering, Dublin City University, Dublin 9, Ireland

**Keywords:** chipless RFID sensors, thermocouple interfacing, voltage sensors, chipless RFID interrogation

## Abstract

To date, no printable chipless Radio Frequency Identification (RFID) sensor-related publications in the current literature discuss the possibility of thermocouple integration, particularly for the use in extreme environments. Furthermore, the effects of a time-dependent stimulus on the scattering parameters of a chipless RFID have never been discussed in the known literature. This work includes a review of possible methods to achieve this goal and the design and characterization of a Barium Strontium Titanate (BST) based VHF/UHF voltage sensing circuit. Proof-of-concept thermocouple integration was attempted, and subsequent testing was performed using a signal generator. These subsequent tests involved applying ramp and sinusoid voltage waveforms to the circuit and the characteristics of these signals are largely extracted from the scattering response. Overall conclusions of this paper are that thermocouple integration into chipless RFID technology is still a significant challenge and further work is needed to identify methods of thermocouple integration. With that being said, the developed circuit shows promise as being capable of being configured into a conventional chipless RFID DC voltage sensor.

## 1. Introduction

This paper sets out to develop and test possible methods capable of interfacing thermocouple and chipless Radio Frequency Identification (RFID) technology. To do this, a Barium Strontium Titanate (BST)-based varactor is integrated into a Very High Frequency/Ultra High Frequency (VHF/UHF) circuit and interfaced to a thermocouple. In addition, the way in which a time-varying stimulus alters the resonant characteristics of a spectral chipless RFID tag is of interest to the authors of this paper. To date, no other works have been found to explore such phenomena and fortunately a DC voltage sensor is a highly suitable device on which to perform such an investigation as the stimulus can be easily controlled.

Many works prior to this have reviewed chipless RFID tags, including that by Herrojo et al. in [1] and the work of these authors in [2]. Both thermocouple and chipless RFID technologies have been printed in-situ using various means, which opens the possibility that a future sensor implementation could be completely printable. The proof-of-concept design is designed and tested in this work and the sensor is also used to investigate how a time-varying stimulus can be appropriately detected from a single dataset. This latter result opens up the possibility that the interrogation requirements of a multi-sensor environment could be reduced by decreasing the interrogation times of each individual sensor and freeing up the reader system to read other sensors in a timely manner. This work makes use of a fully wired RFID tag/circuit as it would lead to a more robust characterization of the sensor than its wireless counterpart.

### 1.1. Current Need for Extreme Environment Compliant Printable DC Voltage Sensors

Usage of sensors in extreme environments brings about a range of design requirements that must be met. Of most interest to this work are the temperature stability and radiation hardness requirements of such a sensor system. Many other works have outlined the unwanted phenomena that occur when using conventional semiconductor technologies under extreme environmental conditions, such as [3,4]. In this context, it can be concluded based on the works presented in References [3,4,5], that only wide bandgap semiconductors offer the necessary protection for operation in extreme environments. Unfortunately, in-situ fabrication of such semiconductor devices into a working RFID Integrated Circuit (IC) would be quite a complex undertaking. Existing chipless RFID tags, however, can readily be fabricated using screen/gravure printing [6], aerosol deposition, and inkjet printing [7,8,9] technologies. Furthermore, highly inert polymers such as polyimide have been printed using these methods [10,11], which could be used as an RFID substrate material.

Aerospace Structural Health Monitoring (SHM) sensor systems frequently need to resist the effects of temperature extremes along with effects caused by ionizing radiation [3,4]. Such systems can experience temperatures from cryogenic levels (−150 °C) to over 1500 °C [12,13,14] and adequate temperature sensing is required in many such structures. For example, the lifetime of a turbine blade can be reduced by as much as 50% if the exhaust gas temperature is not adequately controlled [14]. Such aerospace systems make use of thermocouples [15] and other similar structures make use of fiber optic-based sensors [16,17]. However, there is a clear lack of recent literature on printable methods of integrating thermocouple and chipless RFID technologies. Practical implementations of such a sensor system will of course involve the installation of the thermocouple in the region of interest and the installation of the chipless RFID interface in a location with a more modest, but non-trivial ambient temperature. Such technologies could lead to a significant reduction in sensor installation time and overall weight of the structure [12,13]. Conservative estimates by Dong and Kim in [18] suggest that implementing a full SHM system using wired technology would add an additional 1000 pounds of weight to the craft and would consist of over 10,000 sensors.

### 1.2. Challenges in Integrating Thermocouple and Chipless RFID Technologies

The main issue in integrating a thermocouple into a chipless RFID tag is that it requires the integration of a working DC circuit into the microwave circuit of the RFID tag. To date, no published works have been found that discuss the integration of thermocouple and chipless RFID technology. The only related work is that by Dionne et al. in [19], which uses a varactor diode and a large resistance to respectively convert and isolate a generic DC source from the RFID circuit. Possible reasons as to why thermocouples have not been integrated into chipless RFID designs thus far include the fact that AC signals can readily flow through these devices. Any such current flow through a thermocouple generally results in “self-heating” [20], which would be detrimental to any subsequent temperature measurement. In order to assess how this challenge could be overcome, the following design proposals were considered.

#### 1.2.1. Liquid Crystal Polymers (LCPs)

These materials exhibit dielectric anisotropy [21,22] and their electrical permittivity can be altered in a specific direction, through the application of an electric field in an orthogonal one. The maximum change in permittivity is confined to the normal and tangential permittivity values of the specific polymer. Such a material could be used to convert the DC thermocouple output into a variable capacitance in the microwave circuit. Examples of its implementation in other microwave applications include the works of Liu and Langley in [23] and that by Fritzsch et al. in [24] where the resonant frequency of an antenna is configured to be electrically controllable.

Some issues arise when attempting to use this material in low-voltage applications. The foremost of these problems is the fact that a significant threshold voltage exists in the voltage–permittivity relationship. Conservative estimates of this threshold voltage from [23] exceed 1000 mV. Other challenges associated with the use of these materials is that many of them exhibit frequency-dependent dielectric anisotropy [22] and can also be sensitive to other stimuli such as Volatile Organic Compounds (VOCs) [25].

#### 1.2.2. Electrostatic Actuators

These elements are commonly found in Micro Electromechanical Systems (MEMS) devices and support a variety of actuation uses. The mechanical system comprises two metallic beams that are joined/isolated at one end via an insulative region. Upon application of a potential difference between the two beams, they move towards each other as defined by Coulomb’s law. Resisting this bending motion is the mechanical properties of the insulative layer, which can be modelled as having a stiffness effect, as described by Hooke’s law. As the electrostatic force is an inverse square law and the spring effect is merely an inverse law, the voltage-displacement relationship exhibits a digital-like behavior above a certain applied voltage (pull-in voltage) [26,27]. This voltage can be calculated as the voltage required to reduce the separation by d/3 [27]. Before this digital region in the voltage-displacement curve, an analog region exists where specific voltages lead to specific displacements. This device may be of use to the problem at hand as the work of Thai et al. in [28] demonstrated the use of a bi-metallic strip inside a chipless RFID tag design, as a means of temperature measurement. A similar approach may be possible with a much smaller RFID tag design.

#### 1.2.3. Ferroelectric Materials

Ferroelectric materials, such as those that exhibit a perovskite structure support the formation of an electric dipole within the unit cell. An example of such a material is the ceramic, Barium Strontium Titanate (BST). This material (Ba_x_Sr_1−x_TiO_3_) is a combination of two other materials with perovskite structure, BaTiO_3_ and SrTiO_3_. Upon the application of an electric field, the permittivity of this material is reduced in the opposite plane, and this dependence has been modelled and characterized in numerous works including that by Wexler et al. in [29] and by Wang in [30]. This material has already been used in a host of applications, including its use as a suitable FRAM material [31]. The hysteretic behavior governing the materials use in FRAM technology is temperature dependent and only occurs in the ferroelectric phase of the material. Above the Curie temperature, the paraelectric phase occurs and does not exhibit this behavior [32,33,34]. Operation of the material in this phase also brings about temperature-dependent performance effects [32,34]. The transition temperature between the two phases can be varied by altering the Barium:Strontium (x) concentration [30], but other properties such as loss tangent are also highly dependent on this concentration [35]. Doping can greatly enhance certain properties of the resulting material including loss tangent [36,37], sensitivity [36], and dielectric dispersion [36]. With regard to operating temperatures of this material, works such as that by Nadaud et al. in [33] have shown that deposited BST can exhibit permittivity changes as low as 2% between −75 °C and 100 °C. Furthermore, this work also demonstrates varactor operation at temperatures up to 200 °C. Further efforts are needed in characterizing the performance above this level, however this material can and has been deposited in-situ [31,36,38].

Of critical importance to this discussion is the challenges associated with depositing this material in-situ. Thin film implementations of this material can exhibit significant variations in loss tangent and other similar properties, compared to their bulk counterparts [37]. Likewise, layer thickness can have a significant effect on Curie temperature [39] and sintering requirements must be carefully adhered to, in order to achieve the desired dielectric constant [35]. The work of Pervez et al. in [40] also discusses how the quality factor of the films at low bias voltages is highly dependent on environmental factors in the deposition protocol.

#### 1.2.4. Comparison of Possible Approaches

The following table outlines the merits and shortcomings that each of the design approaches brings. From Table 1, it is evident that the most suitable design approach is to use BST as part of a capacitor design in a chipless RFID tag.

### 1.3. Effects of Dynamic Stimulii on Chipless RFID Interrogation

The presence of dynamic effects in the stimulus being observed could lead to significant levels of distortion of the idealized resonant behavior of the sensor. Since a variable DC voltage source can be easily configured using existing laboratory equipment, one simple way to observe the effects of dynamic stimuli is to apply this signal to the developed RFID sensor.

The dynamic effects investigated include stimulus gradients (ramps) and periodic (sinusoidal) behavior; the latter of which would be quite likely to appear in chipless RFID-based mechanical strain sensing in aerospace SHM applications and other settings besides. To date, no works have been found that discuss the effects of these behaviors on the resonant response of an RFID sensor. This work only considers these effects in relation to the symmetrical bandstop resonant response presented by the Stepped Impedance Resonator (SIR) circuit and thus would not apply to other resonators such as the Electric Field Coupled (ELC) resonator used in [46] or the rhomboid resonator discussed in [7]. Furthermore, the findings of this work are only related to resonators that exhibit purely reactive changes in response to the applied stimulus. This work is not a complete analysis of the problems presented by dynamic stimuli but instead highlights the challenges in accounting for these effects and outlines some possible strategies to overcome these challenges.

The reason that this is seen as important by the authors is that in a large SHM application where thousands of sensors may be in use, a great reduction in sensor interrogation could be achieved if additional, temporal information about the stimulus could be detected from the results of a single interrogation. The analysis performed in this work focusses on the variation in the S21 scattering parameter as this parameter is the parameter most likely to be found in chipless RFID literature as many interrogation systems use a bistatic configuration.

Reader technologies, such as those that use Impulse Radio Ultrawideband (IR-UWB) approaches [47], have shorter interrogation times than those found in swept oscillator -based readers such as conventional Vector Network Analyzers (VNAs). Regardless of this fact, extraction of temporal information is still of interest with both reader systems but to a greater degree with the latter reader type. This work focusses on VNA-based interrogation only, but further work is needed to explore the importance of dynamic stimuli on the measurements performed with the former reader type.

## 2. Methodology

### 2.1. Existing BST Varactors

Although in-situ deposition of BST has already been performed, such operations require specialized equipment and related experience in this area, for the reasons discussed in the previous section. Fortunately, ST Microelectronics supply a range of BST-based varactors (STPTIC Series [45]), which can be used to determine if this material is suitable for the application at hand, without the need for material deposition. This BST-based material, Parascan™ is a doped version of BST and its specified minimum bias voltage is 1V [45]. These capacitors have already been used in other works such as that by Kabir in [43], which biased these varactors at voltages in the sub-one-volt range.

One factor that must be considered when using these varactors in a chipless RFID-based design is that there are only three pins on the devices procured as part of this work, as the ground for both the DC and AC circuits must be the same. This limits the use of certain chipless RFID circuit designs to ones that contain a ground plane.

### 2.2. Test Circuit Design

The integration of impedance sensors into chipless RFID tag designs has been performed in many works prior to this paper. One work of interest is that outlined in [48] by Amin et al. This design utilized a coplanar waveguide design with a λ/4 SIR and used the sensor to ground the final element of the SIR. As this design incorporates a ground plane, it is the ideal design on which to build the final sensor design. The purpose of this paper is merely to characterize BST as a possible candidate for thermocouple integration, thus for simplicity, the λ/4 SIR design used in this work is based on that outlined in Reference [47]. Similar to the design found in Reference [47], this work places the stimulus-sensitive impedance element between the open-circuit side of the SIR and the ground plane. The underlying operation of SIR circuits is described by Pozar in [49]. This circuit design can be readily implemented into a working chipless RFID tag design, as outlined in Reference [2]. This operation is not of critical importance as of yet, as it seems prudent to assess the performance of the circuit independently of as many interfering variables as possible.

The resulting circuit design is described in Figure 1. In this Figure, the underside of the board is shown in yellow and the topside is displayed in green. The standard Coplanar Waveguide was designed to have an impedance of 50 Ω. The SIR used in this design comprises of three cascaded sections of sizes: 0.5 × 22.4 mm, 0.25 × 3 mm and 4.8 × 15 mm. All other relevant dimensions can be found in Figure 1. Two vias are used to allow the varactor to be connected to the ground and to the end of the SIR element, such that both of these elements are not directly connected to each other. A two-pin male header is used to bring the shared ground and bias input to the varactor out of the board for connection to an external DC circuit. On each end of the microwave circuit is an SMA connector to allow for a wired connection to a VNA or for connecting to external antennas. The addition of the vias and varactor add extra inductances and an extra capacitance to the λ/4 SIR design. Similar to the referenced designs, this circuit is designed to exhibit a bandstop characteristic.

### 2.3. Sensor Fabrication

One challenge in the use of the STPTIC range of capacitors in a chipless RFID tag that can be easily fabricated is that the footprint of the varactor is less than 1 mm^2^. As the STPTIC datasheet [45] specifies that the varactor capacitances can vary to a sufficient degree between varactors, two tags are fabricated. This allows for some further comparative analysis between successive devices. Table 2 gives the details of the fabricated sensor and the developed circuit can be seen in Figure 2.

A cheap K-Type thermocouple [50] was acquired from Radionics (RS), which will serve as the temperature sensor. This device is connected into the tag via a two-pin header placed in a bare area of the PCB. Some basic test results of operating the thermocouple at temperatures up to 180 °C can be seen in Figure 3. It is worth noting at this point that many of the printable thermocouples found in the literature [51,52,53] exhibit sub-millivolt output voltages, which are more difficult to acquire using this thermocouple. The ambient temperature recorded during this testing is 19.8 °C.

### 2.4. Circuit Testing

This section outlines how the fabricated device was tested. These tests comprised a mixture of testing with a signal generator and with the thermocouple described in the previous section. As the thermocouple has already been described in the previous section, this section will focus on the test environment and the tests performed.

#### 2.4.1. Temperature Testing and Environment Details

Tests involving the plain thermocouple were performed in both a furnace and in a conventional electric oven. The former had an in-built thermocouple sensor for temperature determination and an infrared thermometer was used to detect the temperature in the latter setting. Since both settings yielded similar, predictable results, most of the subsequent testing was performed using a regular oven. Further testing is also performed on the tags by exposing the circuit to a variety of temperatures and maintaining the bias voltage as a constant. This analysis is performed to assess the degree to which this effect will need to be compensated for and the degree to which bias voltage sensitivity may be a function of the temperature of the circuit. This line of investigation is brought on by the findings of Nadaud et al. in [33] and the temperature sensor tag developed by Mandel et al. in [54], which uses BST as a stimulus-sensitive material.

Figure 4 depicts the test setup used for thermocouple-related tests. This setup comprises the use of an infrared thermometer for the measurement of the ambient temperature of the tag, along with the use of a voltmeter to measure the output of the thermocouple, between successive tests. The furnace contained a digital interface and setpoint control, to allow for the control and detection of the furnace temperature. The thermocouple was positioned in the furnace at the same location as the internal temperature sensor in the furnace, to achieve the greatest possible accuracy during testing. The test procedure involved testing the device at increasing and decreasing furnace temperature setpoints, once the temperature had stabilized. The ambient temperature of the tag was controlled within 1 °C using heat and ventilation sources around the tag. Subsequent testing used a different procedure that involved the collection of fewer frequency sweeps and involved removing the thermocouple out of the furnace before and after testing. This latter procedure allowed for a zero millivolt (reference) measurement and a useful furnace measurement to be performed in a way that both had the approximately same ambient temperature. Testing of the tag at various ambient temperatures between 16 °C and 40 °C were performed using a signal generator as the DC source and by placing the tag in the oven/furnace.

#### 2.4.2. Exploration of Stimulus Gradients

The maximum scan rate of the VNA used in this work is 100 frequency points per second. This opens up the possibility for erroneous readings of the stimulus level, if the stimulus level is a function of time. This problem would of course be less of an issue with a more advanced reader system. However, the fact remains that with any conventional reader that performs a frequency sweep, the spectral signature of the encoded stimulus bit will be altered in proportion to the rate at which the stimulus is changing. This work uses an AimTTi TGF3082 signal generator to perform the relevant ramp and sinusoid generation for the RFID tag. As with the other tests, the NanoVNA V2_2 [55] and associated interface software [56] is used to collect the relevant data. One important point to make is that the Nano VNA V2 is capable of 100 points/s sweeps and can store 201 points locally on the device. Finer resolution is achieved through iterative requests and responses between the PC and VNA, which incur Operating System (OS) and communication overheads. This mode of operation is unsuitable for time-varying stimuli as it appears that the introduced delay will result in distortions of the true dataset. This issue can be avoided by performing sweeps with less than 101 data points. Another issue that must be considered is that it appears that the VNA updates its recorded measurements asynchronously to the operation of the PC GUI and that the GUI simply retrieves the latest data. Thus, the VNA can update the data whilst the GUI application is retrieving the same data, which can lead to small distortions in the data. This can be avoided through successive GUI data requests that are out of frequency/phase with the VNA updates. Figure 5 gives a graphical depiction of the test setup.

Another caveat in the detection of dynamic stimulus effects is that averaging may not be permitted, if significant a-priori knowledge suggests that highly non-linear effects are taking place. This is especially true in the case of sinusoidal effects as successive frequency sweeps will most likely contain sinusoids, which are out of phase with that found in the other sweeps. Another point of relevance to this discussion is the issues related to the ambient temperature sensitivity of the circuit. Although this is a significant problem when the device is being used with millivolt-level bias voltages, as can be seen in the following discussion; this issue appeared far more negligible when the device is experiencing bias voltage changes in the order of several hundred millivolts and when the device is not being operated in the presence of or has a material connection to a nearby heat source, such as a furnace. Furthermore, each test performed in this section comprises of one frequency sweep performed at ambient temperature. Unlike the thermocouple-related tests, variations of ambient temperature between successive tests is of no significant concern to the goals of this section, as there is no need to directly interpret successive tests of the same stimulus.

## 3. Results and Discussion

### 3.1. Large Voltage Biasing of Sensor

The test results presented in Figure 6 reveals that although there are some deviations between the two respective tags, both are sensitive to input voltages as low as 500 mV. One feature of note is that the voltage sensitivity is slightly lower than that assumed by the simulation results. Possible reasons for this include the ambient temperature effects and capacitance tolerances [45] of the STPTIC varactors. From Figure 6 it would appear that the voltage response is largely linear, but it can be seen that this linearity is weaker at voltages below 500 mV. These tests were performed on both fabricated tags and slight deviations do exist in the resonant response. Although this is not of grave concern when operating with voltages up to 2.5 V, they are significant when operating in the millivolt range. It is worth noting at this point that further testing was performed up to higher voltages but the results were in line with the permittivity changes described in the STPTIC datasheet [45].

Further testing at bias voltages up to 4.5 V were performed using a DC power supply and revealed that despite increasing the temperature of the tag itself as seen Figure 7 below, the variations in voltage sensitivity appear largely negligible at those voltage levels. These tests were performed at temperatures up to 40 °C, which did not exceed the thermal ratings of the silk screen on the PCB or the thermal ratings of the VNA cables. Factors involved in the dependence of the resonant frequency and temperature not only include the BST material, but also include the thermal sensitivity of the Rodgers RO4003 substrate [57] and that of the silk screen material.

Of most interest to this work is the thermal sensitivity of the device in comparison to that of its sensitivity to millivolt bias voltages. Initial testing revealed that alterations in ambient temperature of levels around 0.1 °C result in a significant alteration of the resonant frequency, compared to the effect caused by millivolt-level bias voltage changes. Thus, subsequent testing was performed at consistent ambient temperatures to avoid this effect.

### 3.2. Thermocouple Biasing of Sensor

The location of the resonant dip of the thermocouple-based chipless RFID tag is presented in Figure 8 below for two distinct temperatures with the ambient temperature of approximately 19.8 °C. As the levels of noise remained consistent between successive tests and challenges arose in both achieving and maintaining sufficiently accurate temperatures in both the oven and furnace environments, averaging of multiple test results was not performed. This choice was also made because the sampling time of one sensor at this level of resolution was approximately 70 s, including PC communication overheads. Thus, redoing the sweeps would allow sufficient time for the temperature to drop slightly and lead to a small but noticeable measurement error. Furthermore, in a real-life SHM scenario that comprises thousands of sensors, it may not be feasible to successively poll each sensor to achieve adequate averaging of the data. As this section is only focusing on proving that BST can be used to integrate thermocouple and chipless RFID technology, a more robust approach can be taken to detect the true minimum of the curve. Through logarithmic analysis of the frequency sweeps performed, it appeared that a second-order polynomial would be an accurate fit for the data at the bottom of the resonant region. Table 3 gives the R-squared value for the fitting of the second-order polynomial equation to the various datasets. Each of these is quite high, with the lowest (77 °C Curve) having a value of 0.9425. Although this exact approach is only applicable in the context of this design, it does allow for the avoidance of successive polling (thus allowing the reader to interrogate other tags) at the expense of relying on dedicated hardware/software to perform the relevant computations.

Figure 9 depicts the resonant dip location for each of the polynomial curves and it can be seen that a strong linear relationship appears to exist within the data. The major outlier was the 77 °C curve but largely speaking, the response is linear. These test results have been replicated several times but other test results showed a lack of or differing levels of sensitivity to the small bias voltages applied to the tag. Further analysis revealed that these effects largely occurred during fluxuations in ambient temperature of below 0.1 °C.

Based on the discussion above about thermal sensitivity and on the very weak sensitivity of this varactor to millivolt bias voltages, it is the opinion of the authors that this approach is not a suitable approach to achieve thermocouple integration into chipless RFID technology. Further analysis is required to investigate how a stable and consistent measurement performance can be achieved with the existing design. Testing of the device at several different ambient temperatures revealed trendlines in each dataset. These datasets, which mapped thermocouple voltage to resonant frequency, revealed the linear trendlines described in Equations (1)–(3). Figure 10 depicts the plot of these three trendlines for various equivalent temperature inputs.
19.7 °C: MHz = 0.0652 (mV) + 236.11(1)
19.8 °C: MHz = 0.0378 (mV) + 236.07(2)
20 °C: MHz = 0.0179 (mV) + 235.99(3)

The trends in the coefficients of the equations above suggest a slight trend in both the initial resonant frequency and voltage sensitivity tag with respect to ambient temperature. It is important to mention at this point that the minimum R-squared value of the above linear regression analyses was 0.9893. One issue with performing further analysis is the effects of variations in ambient temperature and the differences in rates of heat transfer from the different materials in the sensor and the difference in temperature across the device. This means that the measurement of the ambient temperature is not necessarily indicative of the current temperature of the BST, which is a critical issue as although repeated testing has revealed positive results, the degree to which small variations (<0.1 °C) in BST temperature could be dominating needs to be determined, or indeed driving the sensor response. Furthermore, it cannot be ignored that the Rodgers PCB material and the silk screen material could be heavily involved in the temperature dependence of the circuit performance. This normally would not be an issue for most other sensor designs but the weak electrical sensitivity dictates that this possibility should be considered. This analysis is of critical importance, in determining the magnitude of a possible deadband in the circuit response to weak electric fields, which would mark this design as being unsuitable for this application. Interestingly, the circuit responds differently to millivolt-level bias voltages originating from a signal generator. Further analysis requires testing in a more bespoke environment where the ambient temperature can be accurately controlled and the thermal effects from all relevant sources can be accounted for and/or counteracted against. These thermal sources include the natural ambient temperature, heat transfer from the furnace to the device via the thermocouple leads, and heat generated from the VNA itself. From the equations listed above, it can be seen that a 0.1 °C ambient temperature change is equivalent to a 40–80 kHz change in resonant frequency at a zero-volt bias voltage, whereas the thermocouple-based results in Figure 10 exhibits a sensitivity of 2.9 kHz/°C. This corresponds to a device sensitivity of 0.123% change in resonant frequency for a 100 °C change in temperature. Subsequent testing revealed that the device exhibited millivolt-level voltage sensitivity in both small ascending and small descending temperature gradients. Thus, the electrical effect on the permittivity cannot be completely ignored from the overall behavior of the device but the sensitivity is quite low. Initial results are positive, but the conclusion of this work is that further investigation is required to properly characterize the performance of the circuit, and alternative methods need to be found to achieve thermocouple integration into chipless RFID technology. Of further interest to this discussion is the stability of the measurements taken from this device. All of the tests that were performed under consistent millivolt bias voltages and consistent temperatures revealed that the average deviation between successive equivalent tests was approximately 5.519 kHz, with a standard deviation of 4.406 kHz. This result is critically important as it proves that even within six standard deviations from the mean, the data presented in Figure 9 and Figure 10 could not be generated through other unknown effects causing variations in resonant frequency of the tag. Furthermore, the ambient temperature is only known within 0.1 °C, which corresponds to an approximate change in resonant frequency of 40 kHz, which is the maximum possible value in variation in resonant frequency that is allowed before another stimulus has to be considered as interfering with the resonant frequency of the tag. Considering that the maximum variation that can arise within six standard deviations is lower than this value, this would suggest that the resonant frequency is predominantly controlled by temperature and the applied bias voltage. Thus, in conclusion, further study is required to ensure stable operation of the device, but the sensitivity of any resulting circuit is currently quite low, in comparison to the other non-thermocouple-based chipless RFID temperature sensors reviewed in Reference [2].

### 3.3. Results of Dynamic Stimulii

#### 3.3.1. Exploration of Stimulus Gradient Effects

From the results in Figure 3, it is apparent that the response of the BST varactors to voltages between 1 V and 2 V could be assumed to be approximately linear. Using this range of voltages, several voltage ramps were applied to the tag and the output was measured over a range of frequencies larger than the region that the 1 V and 2 V resonant regions reside in. Note that the duration of these voltage ramps exceeded the time needed to perform the frequency sweep and thus the effects of the transition at the end of each ramp is not present in the collected data. The values of 30 mV/s and 50 mV/s were chosen as the ramp rates to be used as their effects on the resonant region of the device were largest as the latter ramp could reach approximately 90% of its final value within the frequency sweep time. Both positive and negative ramps were performed with starting voltages of 1 V and 2 V, respectively. Figure 11 reveals how a 40-point region around the resonant frequency present in each sweep for each of the ramp rates. Although the detection of the minimum of each sweep was performed manually in this case, the following subsection outlines two methods to detect this using a template dataset.

From the results presented in Figure 11, it can be seen that the negative ramp rates appear to increase the apparent Q-factor whereas the positive ramps decrease it. By calculating the area between the *X*-axis and the various curves, a measure of this change can be determined. This result can be seen in Figure 12.

Before this analysis is concluded, it has to be mentioned that the voltage ramps used in the above analysis had consistent start times with respect to the beginning of the frequency sweep. However, this requirement is only needed if the ramp rate and resonant frequency at timestep zero are sufficient to outrun the frequency sweep or if significant stimulus or frequency dependent losses occur in the sensors operation. Furthermore, these results are a function of other variables besides, including:Sampling rate (Hz).Frequency step size (Hz).Resonator bandwidth.Frequency sweep range.Frequency sweep direction.

The last entry in the above list is quite important as several scenarios can arise in the attempted detection of the spectral signature during stimulus gradients. These scenarios are heavily related to the ramp rate and its initial value at timestep zero of the frequency sweep. Note that it is important to remember at this point that the stimulus-dependent resonant region is assumed to be fixed within a band of frequencies and cannot sit at a frequency below that expected. Furthermore, it is assumed that the frequency sweep covers this entire band and although the sweep could start somewhere else within the band, so as to mitigate against the effects of a large initial ramp value (as described in Scenario 2A), one would need prior knowledge of this magnitude in order to do so. Note that all of the positive voltage sweeps performed in this analysis fall under Scenario 2B. The scenarios are as follows and the definition of (LHS) and (RHS) are given in Figure 13:If the Ramp Direction opposes Sweep Direction:Result: Detection occurs. The minimum point on the curve marks the transition from the sampling of the Left-hand side of the resonant curve (LHS) to the Right-hand side (RHS).If the Ramp Direction does not oppose Sweep Direction:Scenario A: Ramp Rate>Sweep Rate or (Initial Ramp Value + Ramp Rate) > Sweep RateResult: Either no resonant response is detected during sweep as the stimulus magnitude has escaped the band or a setpoint-based resonant region is detected at the maximum stimulus level, if saturation occurs within the sensor system.Scenario B: Ramp Rate < Sweep RateResult: Detection occurs. The minimum point on the curve marks the transition from the sampling of the RHS of the resonant curve to the LHS.

From the scenarios outlined above, it is apparent that certain conditions can occur if the ramp and sweep are in the same direction that can hamper the detection of the resonant region whereas if they oppose each other, these effects do not apply. Of further note to this section and to the subsequent one is that in the case of a single gradient, it is possible to infer further characteristics of the voltage gradient from the results presented in Figure 11. Firstly, by using the setpoint resonant curve as a lookup table or mapping function it is possible to infer the approximate magnitude of the gradient at any of the timesteps within the sweep. This is possible through the results of Figure 12, which can be used to detect the direction of the gradient relative to the direction of the frequency sweep. Based on this fact, it is then known which side of the symmetric resonant curve corresponds to the left- and right-hand sides of the minimum of the ramp-related curves displayed in Figure 11. The ramp signals have been extracted using the procedure outlined above and their values throughout the frequency sweep are plotted in Figure 14. Equation (4) to Equation (7) are the results of linear regression on the aforementioned plots. This procedure is highly dependent on both the resolution of the setpoint and ramp curves in two ways:The resolution of all curves will define the accuracy by which the minimum of the curve(s) is measured.The resolution of both ramp and setpoint curves will define the accuracy of the inferred voltage at each timestep in the frequency sweep, as even with the addition of linear interpolation, errors will still exist.

Also, of significant note to this discussion is the relevance of the minimum of the curve with respect to the instantaneous voltage applied to the tag. As the S21 magnitude of the ramp sweep approaches the minimum value of that of the setpoint sweep, one useful observation can be made. Assuming that stimulus- and frequency-dependent losses do not occur in the sensor system (reactive impedance changes only), the point at which the minimum of the ramp curve equals that of the setpoint curve represents the resonant frequency of the instantaneous magnitude of the stimulus at that timestep in the frequency sweep, as no other S21 magnitude can exceed this value. Thus, the correct interpretation of the calculated minimum is that it represents the assumed crossover of the instantaneous ramp voltage resonant frequency and the continuous frequency sweep.
500 mV/s Trendline*: V =* 0.0052(*f) +* 0.2635(4)
−500 mV/s Trendline*: V =* −0.0056(*f)* + 2.9452(5)
250 mV/s Trendline*: V =* 0.0025(*f)* + 0.5504(6)
−250 mV/s Trendline*: V =* −0.0029(*f)* + 2.5876(7)

From the results presented in the Equations supplied above, it is possible to infer the ramp rate in mV/s units. This can be done based on the fact that the NanoVNA V2_2 is capable of 100 samples/s, thus given the original datasets consist of 100 points. It can be concluded that each frequency sweep step in the above Figure corresponds to 0.0052 V, −0.0056 V, 0.0025 V, and −0.0029 V ramp change, respectively. Therefore, a 100-point sweep that takes one second to complete results in a change of 0.52 V, −0.56 V, 0.25 V, and −0.29 V, respectively. Therefore, it can be concluded that the ramp rates extracted from the data are 520 mV/s, −560 mV/s, 250 mV/s, and −290 mV/s. Although errors exist in all extracted ramps except the 250mV/s test, it has to be noted at this point that several factors influenced all of these results, including:No linear or other form of interpolation was performed on the lookup of the true resonant point on the setpoint curve; thus, the accuracy is totally at the mercy of the setpoint curve resolution.Even with interpolation, the setpoint curve had a step size of 1.5 MHz which is quite significant. It is worth noting that the greatest errors induced by this factor will be around the results of the lookup at positions around the minimum value and this poor resolution also hampers the accurate detection of the minimum of the setpoint curve.All other sweeps performed consisted of only 100 points, which led to a resolution of 0.9 MHz/step.The voltage value that has been inferred is based on the accuracy of the linear relationship between resonant frequency and voltage, displayed in Figure 3.

#### 3.3.2. Exploration of Sinusoidal Stimulus Effects

Testing with a sinusoidal stimulus was performed with frequencies from three to ten Hertz.

Figure 14 and Figure 15 reveal the magnitude of the scattering parameters for several five Hertz tests with a peak-to-peak amplitude of 400 mV and a DC bias of 2 V. As can be seen in the Figure 15 and Figure 16 below, the characteristics of the scattering parameters are highly affected by sinusoidal excitation.

Of most importance here is the distinction between this and the earlier ramped stimulus, the latter of which will not consist of multiple possible true minima in the S21 data as the resonant frequency of the tag cycles up and down the frequency range and coincides with the instantaneous measurement. This behavior poses an interesting question; at any point in the resulting data, which side of the resonant region is being observed? A diagram of the idealized resonant S21 response is given in Figure 13. In this figure, it can be seen that two possible frequencies correspond to the same S21 magnitude. Therefore, the direct use of this curve as a mapping function (continuous form) or lookup table (discrete form) requires further inferences in order for the result to be deemed valid.

If the resonant bandstop S21 response is captured, it can be used to convert the measured magnitude at each frequency to the actual resonant frequency at that timestep in the sweep. There will of course be two curves generated using this lookup function as it cannot yet be discerned which parts of either curve is valid. For the subsequent analysis, a 41-point curve (20 points each side of the resonant frequency) was used as the lookup function. By taking all the magnitudes in a 5 Hz sinusoid with an amplitude range of 1000–2450 mV and mapping them to their distance from the resonant frequency, the following curves were generated. Note that the sweep performed encapsulated the resonant regions of the maximum and minimum voltage of the sinusoid. These curves (shown in Figure 17) contain less than 100 points as not all points in the S21 data can be mapped to equivalent resonant frequencies as the frequency window used to generate the lookup function will be no wider than the resonant region itself. In Figure 17, there are two curves, one for both the ascending (RHS) and descending (LHS) parts of the resonant curve seen in Figure 13a above.

Similar to the minimum of the ramp data discussed earlier, the points at which the two datasets converge signify points in the frequency sweep where the minimum of the instantaneous resonant region coincides with the interrogation/sweep frequency. These frequency values are true measurements of the instantaneous voltage that can be directly mapped back to that value through an existing frequency–stimulus equation. From the results in Figure 17 above, it can be seen that several successive points have values of zero, which is caused by the resolution of the lookup function, which will exhibit diminished sensitivity at regions of greatest change such as the bottom of the resonant region. This function had a resolution of 1 MHz and no interpolation was performed for the sake of simplicity. What is known at each of the converging points between the two curves is that it signifies the cross-over of the moving resonant region with the instantaneous sweep frequency. Thus, it is known that a transition occurs after each of these converging points in that the valid side of the mapping function is reversed, i.e., the next sweep frequency should use the opposite side of the resonant curve in Figure 13a as its lookup. There are several possible ways to determine which side of the lookup curve should be used on a particular side of the convergence points shown in Figure 13a. If the reader has the phase information of the S21 parameter, this provides an adequate method to discern the correct lookup procedure. Figure 13b above gives the idealized phase response of the setpoint-generated resonant response, where a transition occurs from positive to negative phase at the minimum of the resonant curve. Figure 18 reveals the S21 phase response for the data used to generate the curves in Figure 16. Of most significance here is the fact that the phase transitions do not exhibit the same plateaus seen in Figure 17 and thus the approximate transition point can be determined from this dataset. The requirement for phase information is not an issue for a VNA but it does mean that a simpler, Scalar Network Analyzer is not as well suited to collecting the data for the task at hand. However, one key point of note here is that the resolution of the relevant curves would undoubtedly fix this issue and the results in Figure 17 can be used to roughly determine the transitions as the corresponding resonant frequency will be approximately correct regardless. Subsequent testing made use of linear interpolation with the lookup curve data and resulted in a significant improvement in the interpretability of the types of curve generated in Figure 17. The main point of note here is that the plateaus present in Figure 17 will disappear and a parabolic-like curve will take their place, thus signifying that two transitions occur between the LHS:RHS pair and not just one.

The requirement for phase information to complete the lookup procedure can be avoided through more simplistic methods that require just the scalar S21 data. One option is to plot both possible stimulus curves and with a-priori knowledge can infer which curve is valid. Figure 19 shows the results of generating the two possible curves for a five Hertz signal with an amplitude between 1800 mV and 2200 mV. From this curve, it can be seen that the “Option#2” trace is the valid curve as the other curve diverges off in a large gradient and other such curves diverged into assumed negative values, which is impossible as it implies that the stimulus resonant region has left its assumed band of frequencies. This method, although suitable in some circumstances, may not exhibit the required characteristics in the invalid trace to allow for its rejection. Other methods to detect the correct side of the resonant curve is being measured include comparing the area under the curve of the measured S21 step and that of the lookup function or in the case of symmetric resonant behavior, comparing the area on both sides of the minimum determined earlier. These methods would be haphazard however as they would be highly sensitive to the resolution of all relevant frequency sweeps and highly sensitive to noise. With regard to the preceding section on stimulus gradients, either of these approaches can be used to detect the true minimum of those curves. Alternative methods such as finding the minimum difference between the ramp curve integral and that of the setpoint curve do not require a lookup function or phase information but there may exist many corner conditions in which this method will fail.

Based on the results of the above analysis, the data for following Figure was generated. This is the extracted 10 Hz sinusoid with an amplitude of between 1900 mV and 2100 mV. The average amplitude of the trace in Figure 20 is slightly higher than expected as the resolution of the steps of the lookup function result in a 44 mV change in output voltage per step and the lookup frequency is determined as the first S21 magnitude to exceed the lookup value on either side of the curve. Subtraction of this small data manipulation-induced bias error in relevant locations reveals that the measured voltage does not exceed the applied values. Of most interest on this curve is that the sinusoid appears to have some small ramp in its DC offset and small errors in the amplitude of the sinusoid. Reasons why this occur involve the following behaviors of the system:The frequency sweep can only accurately determine the peak voltage towards the end of the frequency sweep as the bandwidth of the resonant region and that of the lookup function will not always be comparable with the change in resonant frequency caused by an amplitude of this size. A similar effect occurs at the beginning of the sweep with the minimum amplitude of the sinusoid.The frequency sweep and that used to generate the mapping function has a finite resolution.The spacing between successive points in a discrete setpoint resonant region for a fixed change in S21 magnitude vary nonlinearly in accordance with the nonlinear characteristics of the bandstop response curve.The portions of the S21 magnitude outside of the resonant region are not perfectly flat and may have gradients in the direction of this region. This would cause an invalid use of the lookup function at those frequency points.Small variations in ambient temperature may exist between this test and that used to develop the voltage-frequency lookup and the impedance behavior of the SIR circuit to a varying bias voltage may not be purely reactive.

Despite these failings in the extraction of the sinusoid, the period can be estimated as the frequency steps in the sweep each correspond to approximately 10 ms in duration and thus each period of the sinusoid in Figure 20 should contain, on average, 10 measurement points. Further analysis that utilized linear regression between data points on the lookup curve yielded similar behavior to that seen in Figure 20 below but with the amplitude of the sinusoid largely bounded between 1900 mV and 2150 mV throughout the frequency sweep.

Further test results (no interpolation) are shown in Figure 21 of a 5 Hz signal with a DC bias of 2 V and a peak-to-peak amplitude of 400 mV. Consistent results are generated from the subsequent analysis that, although is incomplete, does successfully extract information such as approximate amplitude and frequency.

As mentioned in earlier sections, linear interpolation of the lookup function data can improve voltage measurement accuracy. Although this interpolation is not quite a valid means to improve lookup accuracy, it will reduce the resolution of the resulting voltage traces. Further testing of 3 Hz input signals revealed complications in the extraction of the instantaneous voltages around the LHS-RHS transition region in the curve. These artefacts were quite significant, but the usage of interpolation greatly improved the extracted voltage signal. Figure 22 is a graph of the interpolated sinusoid and the spurious artefact in the center of the figure is relatively small, compared to that seen in Figure 23, which is the original extracted sinusoid determined without interpolation. This effect took place to larger degrees in lower frequency signals and at points in the LHS-RHS curves where they converge. The reason that this artefact shows up in the extracted traces to greater degrees in low-frequency sinusoids is caused by the sensitivity of the voltage value extractions around this minimum point is due to the shape of the bandstop curve and the accuracy of the determination of its minimum value.

## 4. Conclusions

### 4.1. Results of This Work

Overall, the main goal of this work was to investigate methods to achieve integration of chipless RFID and thermocouple technologies. The results presented in the above sections reveal that thermocouple integration into chipless RFID is a significant challenge. Although further work is needed to further develop the sensor, this work does indeed outline the challenges posed by this problem and investigates the best-known possible solutions. Overall, the design tested above is capable of operation with larger input bias voltages and could find uses in other application settings.

Testing of the sensor with stimulus gradients and a sinusoidal stimulus reveals that these effects have a significant effect on the spectral signature of the tag. Furthermore, it is possible to infer the temporal characteristics of the original stimulus using basic computational techniques.

### 4.2. Future Goals

Although some useful conclusions can be drawn from the results of this paper, there are many unexplored research questions relating to both chipless RFID DC voltage sensors in extreme environments and to the possibility that arbitrary dynamic stimuli can be successfully detected in a single frequency sweep. Of most concern are the following research questions:How can the electrical sensitivity of this device be enhanced, and its thermal sensitivity lowered?What other methods exist that would allow possible successful thermocouple integration into chipless RFID?What are the effects of ionizing radiation and general aging on the performance of this device?Can a combination of materials such as BST, Polyimide, and a metallic conductor be deposited together into a composite sensor in a sequential fashion onto generic substrates in a timely fashion?How well can the dynamic stimulus extraction methodology above perform in a wireless test?

Further testing will focus on testing the device in a more controlled environment and developing methods to enhancing then sensitivity of the device and, if necessary, a sensitive and accurate method for measuring ambient temperature.

## Figures and Tables

**Figure 1 micromachines-11-01019-f001:**
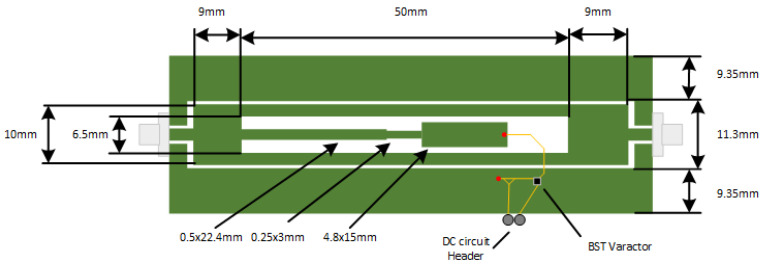
Circuit layout.

**Figure 2 micromachines-11-01019-f002:**
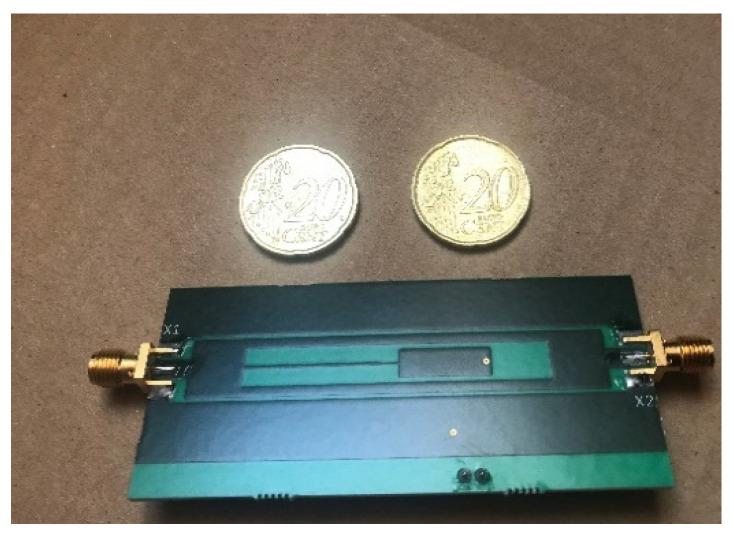
Developed Tag.

**Figure 3 micromachines-11-01019-f003:**
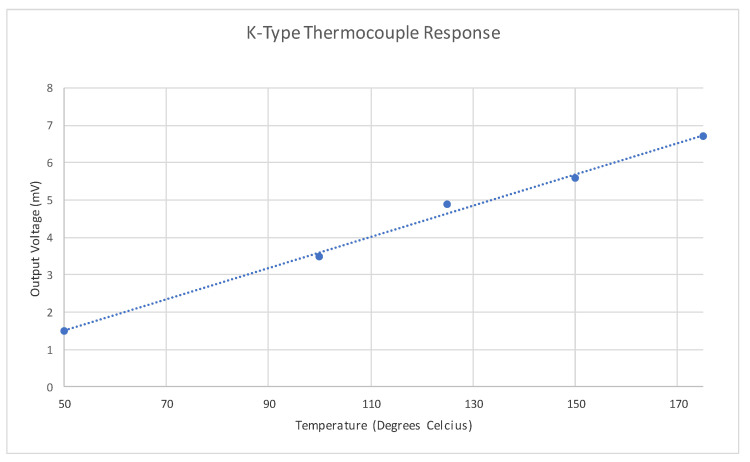
Thermocouple response.

**Figure 4 micromachines-11-01019-f004:**
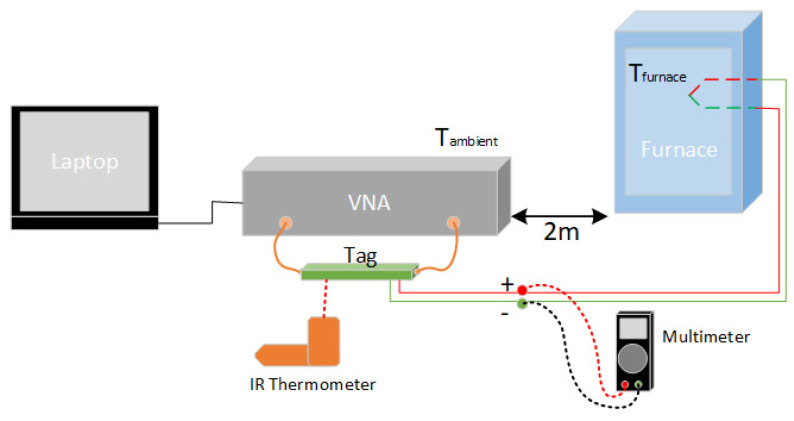
Thermocouple interfaced circuit test setup.

**Figure 5 micromachines-11-01019-f005:**
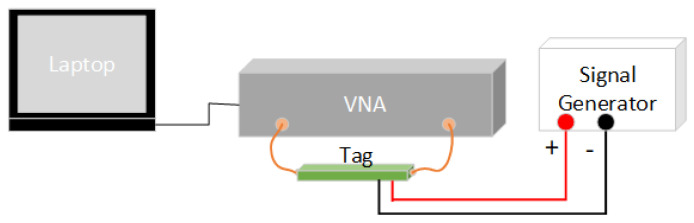
Test setup with signal generator.

**Figure 6 micromachines-11-01019-f006:**
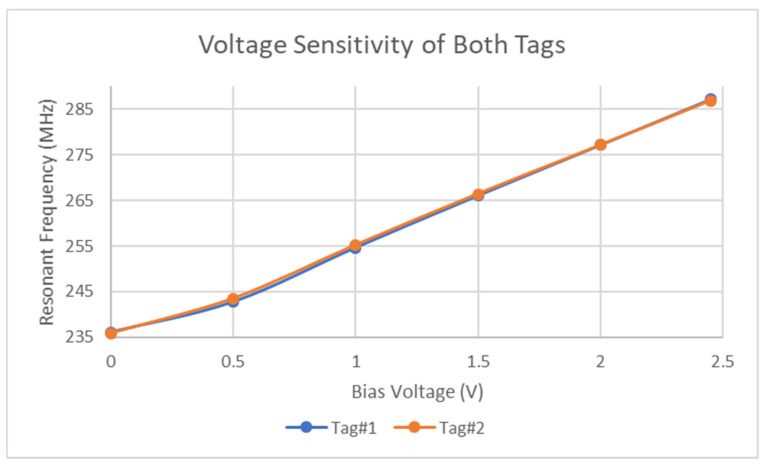
Large voltage sensitivity of tag.

**Figure 7 micromachines-11-01019-f007:**
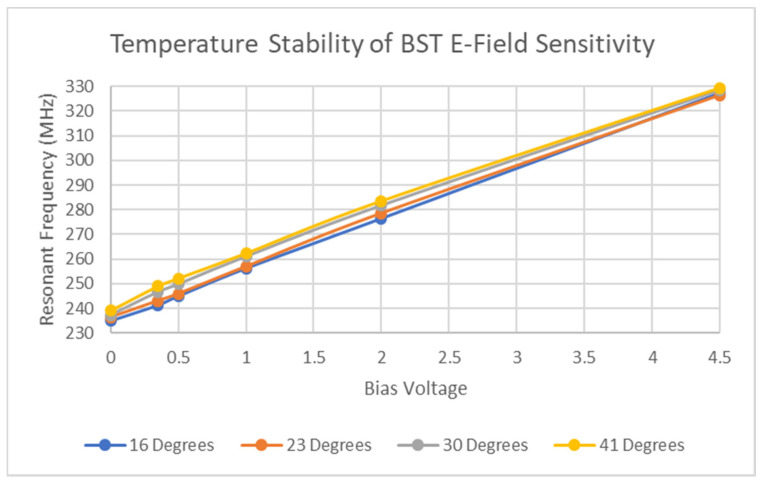
Temperature analysis of tag sensitivity.

**Figure 8 micromachines-11-01019-f008:**
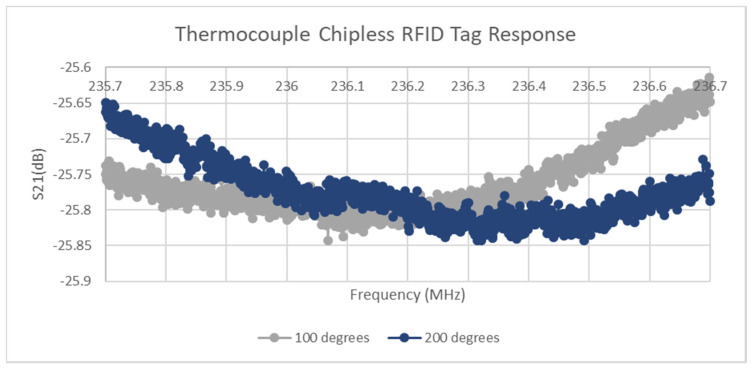
Thermocouple-interfaced sensor response.

**Figure 9 micromachines-11-01019-f009:**
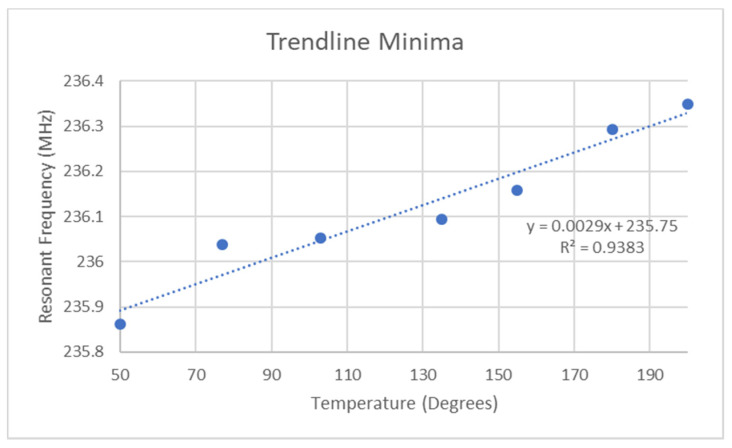
Best Temperature response from sensor.

**Figure 10 micromachines-11-01019-f010:**
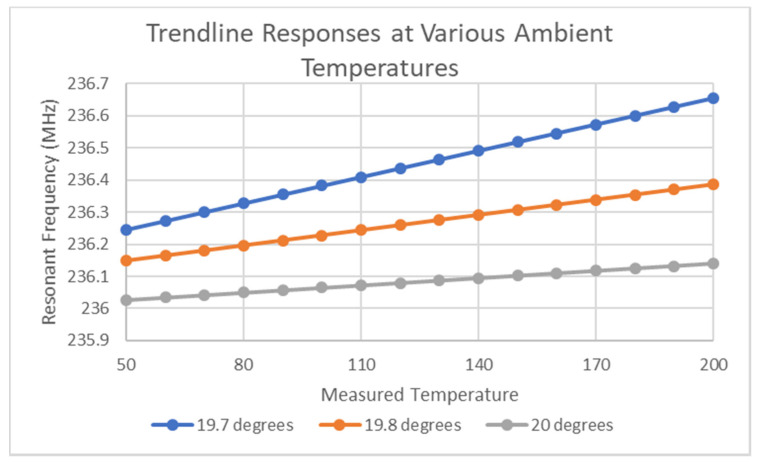
Trendline plots for various temperatures.

**Figure 11 micromachines-11-01019-f011:**
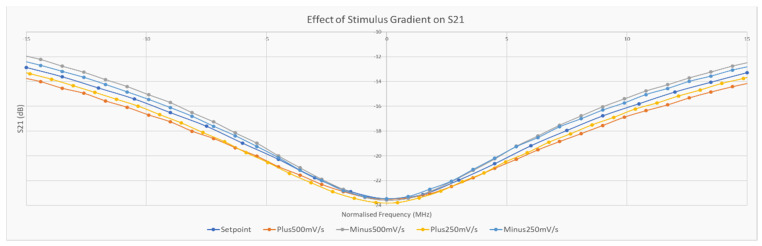
Stimulus gradient results.

**Figure 12 micromachines-11-01019-f012:**
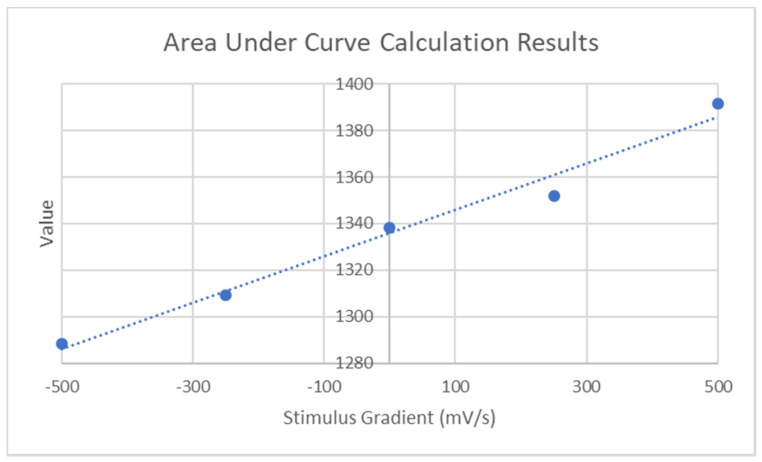
Relationship between ramp rate and area under curve.

**Figure 13 micromachines-11-01019-f013:**
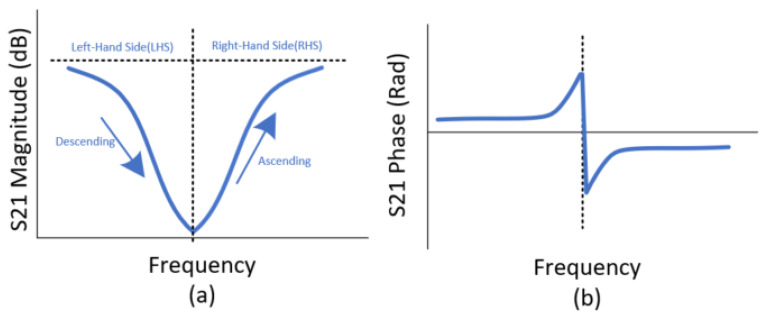
Idealized S21 response of resonant region; (**a**) Magnitude Response, (**b**) Phase Response.

**Figure 14 micromachines-11-01019-f014:**
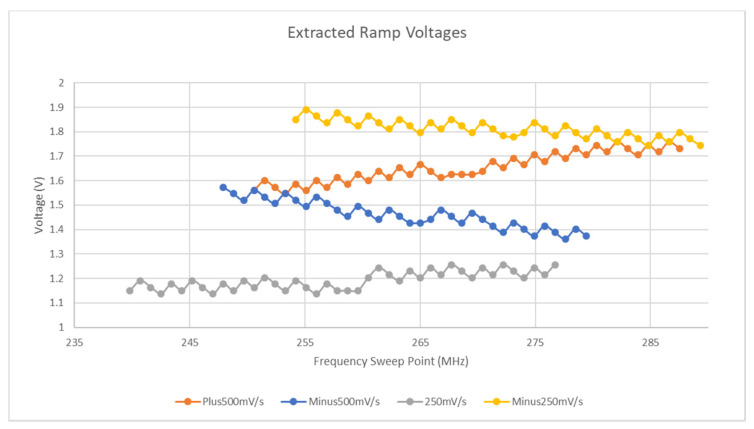
Extracted Ramp Voltages.

**Figure 15 micromachines-11-01019-f015:**
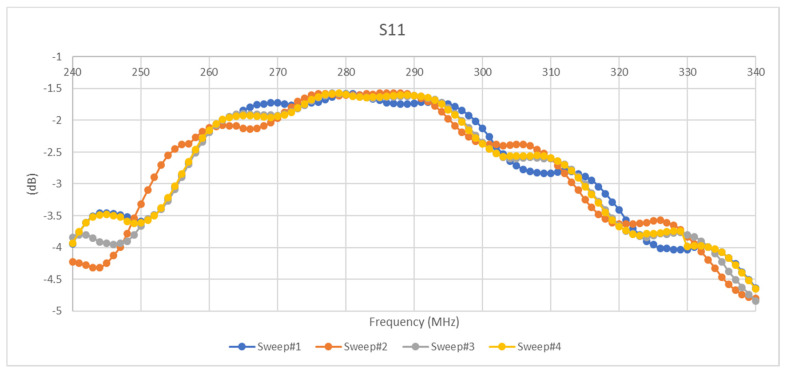
S11 results of sinusoidal excitation.

**Figure 16 micromachines-11-01019-f016:**
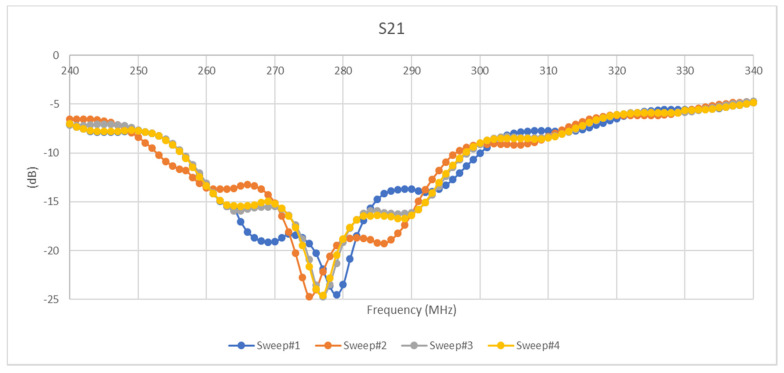
S21 results of sinusoidal excitation.

**Figure 17 micromachines-11-01019-f017:**
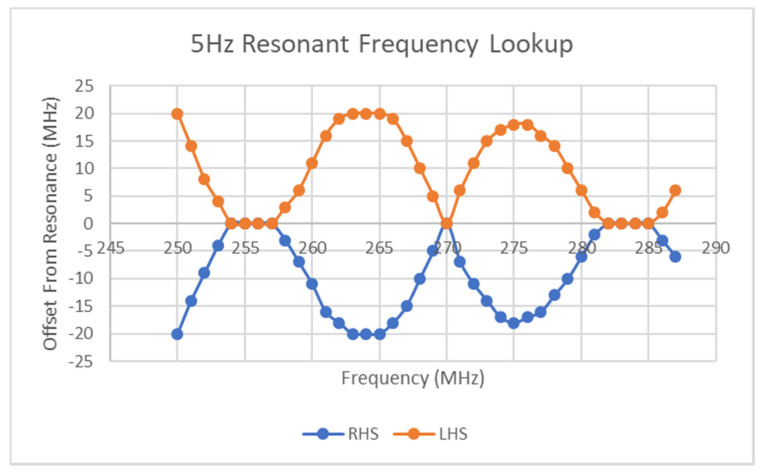
Result of the use of both sides of resonant lookup curve.

**Figure 18 micromachines-11-01019-f018:**
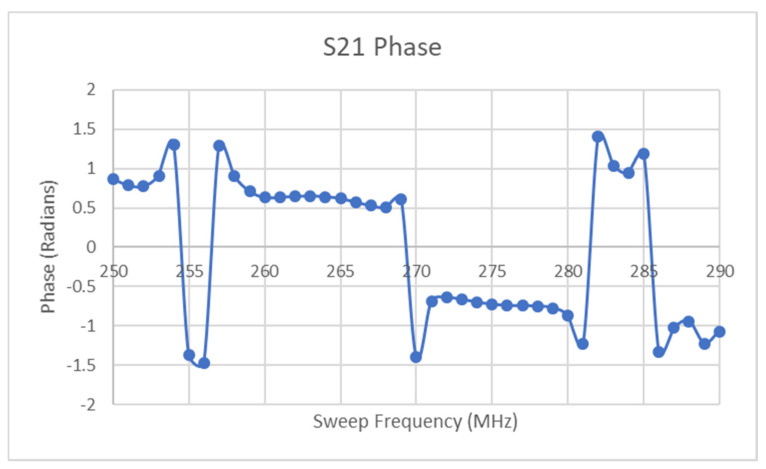
Phase response of S21 data.

**Figure 19 micromachines-11-01019-f019:**
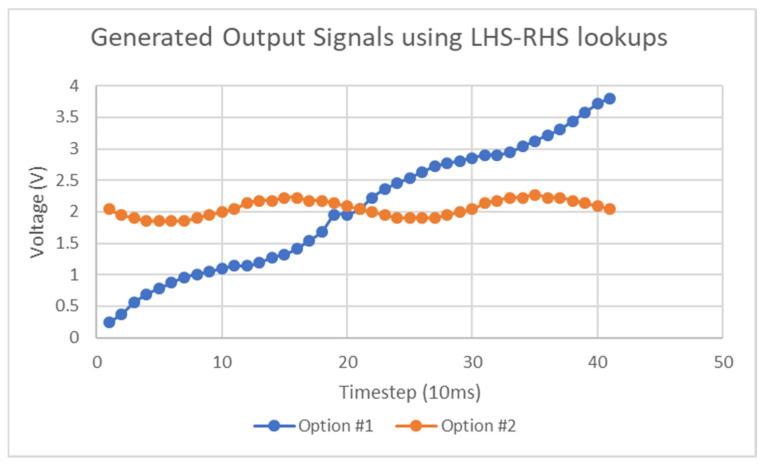
Possible stimulus curves applied to tag.

**Figure 20 micromachines-11-01019-f020:**
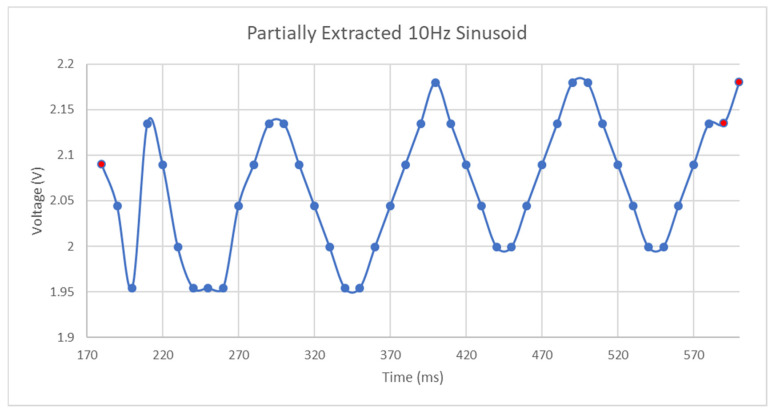
10 Hz partially extracted sinusoid.

**Figure 21 micromachines-11-01019-f021:**
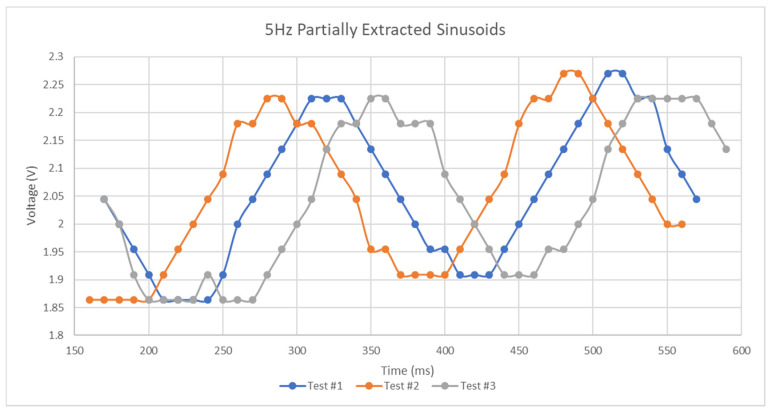
5 Hz sinusoid extractions.

**Figure 22 micromachines-11-01019-f022:**
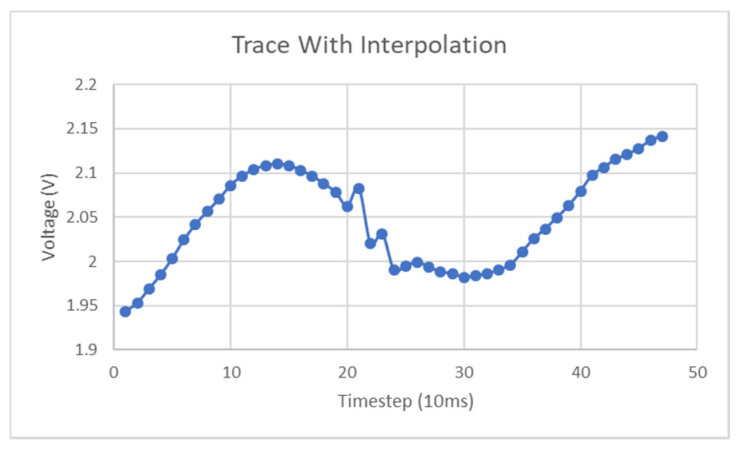
3 Hz interpolated lookup.

**Figure 23 micromachines-11-01019-f023:**
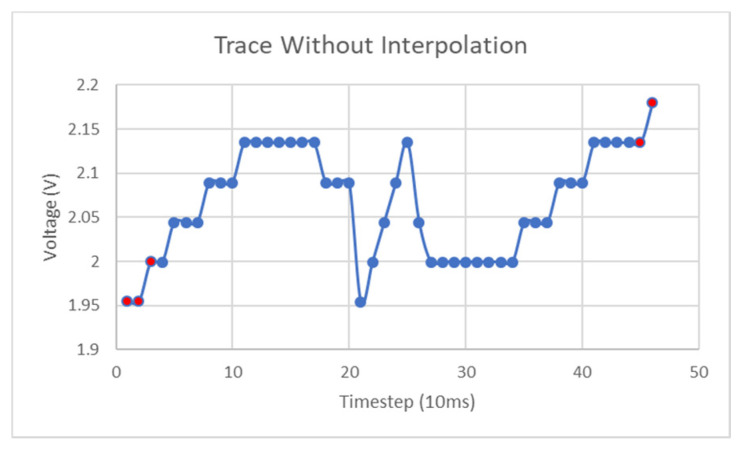
3 Hz first index lookup.

**Table 1 micromachines-11-01019-t001:** Comparative analysis of varactor materials.

Method	Merits	Shortcomings
LCP	Have been successfully used in other microwave-related applications [23,24] Can be deposited in-situ using Chemical Vapor Deposition(CVD) [41]	Significant threshold voltage exists [23] Sensitive to other stimuli [25,42] May not be suitable for high pressure/temperature environments [42]
Electrostatic Actuator	Behavior has been well explored and documented [26,27] Less complex capacitor design	Resulting RFID design will be very small and require very high frequencies of interrogation Serious fabrication challenges Hysteretic behavior also exists
BST	May support mV input voltages [43,44] Has been deposited via inkjet technology [38] Operation has been verified at relatively high temperatures Has high input impedance [45]	Cannot discern +/− voltage inputs due to symmetry in ε-V curve [45] Sensitive to temperature [32,33] Fabrication is still a significant challenge Unknown (supposed low) sensitivity to mV inputs, based on mathematical model [43,44] Hysteresis can occur under certain conditions

**Table 2 micromachines-11-01019-t002:** Sensor design specifications.

Specifications
Dimensions: 82 mm × 36 mm (excluding connectors) PCB Material: Rodgers RO4003 Resonant Region(s): 235 MHz and 1090 MHz upwards DC Voltage Range: 1–24 V [45]

**Table 3 micromachines-11-01019-t003:** Curve fitting accuracy of resonant response.

Temperature (°C)	Polynomial R^2^ Value
50	0.9713
77	0.9425
100	0.9846
135	0.9784
155	0.9895
180	0.9969
200	0.9948

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
