# Peer review of "Current Progress towards the Integration of Thermocouple and Chipless RFID Technologies and the Sensing of a Dynamic Stimulus"

_micromachines, 2020, doi:10.3390/mi11111019_

Round 1
Reviewer 1 Report
The paper presents a proof-of-concept of a chipless RFID DC voltage sensor capable of interfacing to a thermocouple. The results are properly discussed and the necessary future work is outlined. The paper is well written and according my opinion it can bring an interesting look in the area of chipless RFID.
Author Response
Overall, we(the authors) would like to thank the reviewers for their constructive feedback and suggestions and would welcome any further recommendations you may have. The text changes made to the manuscript are seen in this document and the manuscript in blue.
Reviewer 2 Report
.) Chapter 2.2ff: A detailed description of the sensor circuit is missing. Provide a detailed outline of the antenna and the circuit varactor + thermocouple. Also the measurement circuit including the VNA and the signal generator should be sketched.
.) Fig. 6: From repeated measurements a statistical analysis of the measured data should be provided to show the mean and std values of the regression line. Also the 95% confidence should be analyzed to show the performance of the sensor principle.
.) The proposed device is called a RFID sensor including a thermocouple. Clear missing in the study: Field tests of the device using an interrogation unit which provides the RF signal and detects the response of the sensor. A detailed discussion of the results from the field tests. Only VNA measurements are clearly too few for a study on a wireless RFID sensor.
Round 2
Reviewer 2 Report
The paper is now fine and ready for publication